# Antagonist Temperature Variation Affects the Photosynthetic Parameters and Secondary Metabolites of *Ocimum basilicum* L. and *Salvia officinalis* L.

**DOI:** 10.3390/plants11141806

**Published:** 2022-07-08

**Authors:** Lucian Copolovici, Dana M. Copolovici, Cristian Moisa, Andreea Lupitu

**Affiliations:** Faculty of Food Engineering, Tourism and Environmental Protection, Institute for Research, Development and Innovation in Technical and Natural Sciences, Aurel Vlaicu University, Elena Dragoi St. No. 2, 310330 Arad, Romania; lucian.copolovici@uav.ro (L.C.); dana.copolovici@uav.ro (D.M.C.); moisa.cristian@yahoo.com (C.M.)

**Keywords:** cold and heat stress, basil, sage, green leaf volatiles, volatile organic compounds, photosynthesis, total phenols, flavonoids

## Abstract

Climate change is one of the main challenges for actual and future generations. Global warming affects plants and animals and is responsible for considerable crop loss. This study studied the influence of antagonist successive stresses, cold–heat and heat–cold, on two medicinal plants *Ocimum basilicum* L. and *Salvia officinalis* L. The photosynthetic parameters decreased for plants under the variation of subsequent stress. Net assimilation rates and stomatal conductance to water vapor are more affected in the case of plants under cold–heat consecutive stress than heat–cold successive stress. Emissions of volatile organic compounds have been enhanced for plants under successive stress when compared with control plants. Chlorophyll concentrations for plants under successive stress decreased for basil and sage plants. The total phenolic and flavonoid contents were not affected by the successive stresses when compared with the plants under only one type of treatment.

## 1. Introduction

Climate change is a global situation affecting every nation and disturbing the world’s economy [1,2]. During the 21st century, the global temperature has increased by more than 3 °C [3,4,5]. Additionally, weather patterns are disrupted [6,7], and possible sudden major climate shifts have been predicted [8,9].

Plants are very vulnerable to climate change effects [10,11], such as changes in temperature [1,2,5], water status (drought, flooding) [12], rising atmospheric CO_2_ concentration [13,14,15], the attacks of herbivores [16,17,18,19] and pathogens [16,20,21]. Among these climate change effects, increased temperatures negatively impact plant development and crop production [1,11,22]. Recent climate change scenarios forecast both increasing average temperatures and an increasing frequency of heatwaves [5]. The heatwaves caused by global climate change might affect the plant’s capacity to adapt to thermal stress [1,11,23]. For most plant species, the ideal growth temperature varies between 20 °C and 30 °C [24,25,26]. Temperature stress for plants can be considered to be between ten and fifteen degrees above or below the ideal growth temperature [23,26]. However, the severity of temperature stress depends both on the temperature above and below the ideal growth temperature and the period of the applied temperature. The plants most susceptible to climate change are expected to be the species with low thermotolerance [1,23,27]. Basil (*Ocimum basilicum* L.) and sage (*Salvia officinalis*) (*Labiatae*/*Lamiaceae* family) are popular culinary and medicinal herbs and are moderately temperature-tolerant plants grown in temperate regions [28,29,30]. 

The effects of heat stress on plants can be defined as a decrease in photosynthesis, Rubisco activity, and the synthesis of reactive oxygen species (ROS) in analyzed plants [26,30,31]. The reduction of photosynthesis in plants subjected to heat stress is linked with a decrease in photosystem centers II (PSII) activities and photosynthetic electron transport activity [31,32]. 

On the other hand, low-temperature stress (chilling) is undergone by plants from temperate regions and is faced by plants when temperatures fall below 10 °C [33]. The effects of chilling on plants can include s loss of vigor, cellular dehydration, wilting, chlorosis, sterility, and membrane injury attributable to ice formation [25,33]. Apart from photosynthesis and cellular modification, plants subjected to temperature stress suffer alterations in secondary metabolism, including changes in volatile organic compound (VOC) emissions [34,35,36]. Under ideal growth parameters, some plants emit VOCs, such as isoprene, mono-and sesquiterpenes, and oxygenated volatile compounds [36,37]. Under stress conditions, the emission of VOCs depends on the severity and the duration of stress applied. Even minor modifications in the growth or exposure temperature lead to changes in the level and pattern of VOCs emissions [34,37,38].

Alongside VOCs, another important biochemical defense mechanism for temperature stress protection is the enhanced production of secondary compounds (such as phenolic compounds or flavonoids). These phenolic compounds and flavonoids are two essential groups of secondary plant metabolites that protect plants against changes in temperature regime through their antioxidant properties, which can remove ROS [39,40]. Changes in growth temperature parameters cause an increased production of phenolics and flavonoids in plants [41,42].

Basil is an annual aromatic plant, and sage is an evergreen subshrub that is widely grown because of its pleasant odor and taste. Basil and sage plants are two essential medicinal plants, widely used for their extraordinary properties, including antihidrotic, spasmolytic, antiseptic, and anti-inflammatory effects. Sage can also treat mental and nervous conditions [12,43]. Besides those remarkable properties, the plants also have economic importance in several Mediterranean and European countries [44]. Those plants are grown annually for fresh and processed consumption. For example, sage is produced only in Europe at approximatively 1200 ha, and basil at 600 ha [44]. Tropical climates are the places that basil plants prefer. The plants thrive in warm conditions; basil can grow in temperatures as high as 30 °C and grows extraordinarily well up to that temperature [45]. The plants grown in the field are often exposed to higher temperatures than the optimal growth temperature. In the field environments, due to sun flecks, relatively brief heat periods during which leaf temperatures can exceed air temperatures by 10 °C are common [46,47]. In tropical agricultural species, the known limits of tolerance to short-term increases in leaf temperature can range from 35 to 54 °C [48].

Despite their agricultural importance, how basil and sage’s photosynthetic characteristics respond to temperature stress is limited. To our knowledge, no information is available on how VOCs’ production of basil and sage leaves is modified by temperature stress. This is a significant omission, given that more frequent and intense heat periods are coming in the future due to global warming and possible sudden major climate shifts. Understanding how climate can change and predicting how plants will respond, especially to temperature impacts, is critical for plant breeders to ensure global food security. Temperature stress tolerance (tolerance to second powerful stress following different antagonist stress) has attracted interest as a potential technique for evaluating stress-resistant plants to contribute to global food security.

Our study aimed to evaluate the impacts of different temperatures and the effects of high temperature–low temperature, and low temperature–high temperature on basil and sage plants. After applying the variation of temperature, the photosynthetic parameters, volatile organic compounds (VOC), chlorophyll content, flavonoids, and phenolic compounds were estimated to vary.

## 2. Results

### 2.1. The Effects of the Variation of Temperatures on Photosynthetic Characteristics

The net assimilation rate of control basil plants (subjected only to temperature variation) decreased when 25 °C treatment was applied, and decreased from 2.90 to 0.20 µmol m^−2^ s^−1^ when 60 °C treatment was applied. As can be seen in Figure 1b, for the control sage plant, a slight increase can be observed from 3.80 (25 °C) to 3.98 (30 °C) µmol m^−2^ s^−1,^ followed by a drastic decrease to 0.05 (60 °C) µmol m^−2^ s^−1^ (Figure 1).

In the case of plants subjected to both treatments, variations of temperature (heat) followed by cold (4 °C), or vice versa, decreased the net assimilation rate. In the case of basil plants subjected to cold followed by the variation of temperature, the net assimilation rate decreased from 1.70 (25 °C) to 0.19 (60 °C) µmol m^−2^ s^−1^. In the case of basil plants subjected to the variation of temperature followed by cold, a decrease can be observed from 1.65 (25 °C) to 0.12 (60 °C) µmol m^−2^ s^−1^. As shown in Figure 1, the basil plants subjected only to temperature variation had a higher net assimilation rate than plants subjected to the second treatment. In the case of plants treated only with 25 °C, the net assimilation rate is higher than plants treated to 25 °C followed by 4 °C.

For basil plants, as can be seen in Figure 2a, the stomatal conductance to water vapor (g_s_) had a slight increase for plants subjected only to the variation of temperature from 19.2 (25 °C) to 20.5 (35 °C), followed by a decrease to 5.4 (60 °C) mmol m^−2^ s^−1^. The same pattern was observed for basil plants exposed to cold followed by the variation of temperature: a slight increase in g_s_ was observed from 11.9 (25 °C) to 11.3 (30 °C) mmol m^−2^ s^−1^. In the case of basil plants exposed to the variation of temperature followed by cold (4 °C), a decrease in g_s_ was noticed from 16.03 (25 °C) to 1.10 (60 °C) mmol m^−2^ s^−1^.

Regarding the sage plants exposed only to temperature variation, a slight increase in g_s_ can be perceived from 23.02 (25 °C) to 3.35 (60 °C) mmol m^−2^ s^−1^. Sage plants subjected to 4 °C followed by the variation of temperature had a slight increase in g_s_ from 8.92 (25 °C) to 15.39 (45 °C) mmol m^−2^ s^−1^, followed by a decrease to 2.47 (60 °C) mmol m^−2^ s^−1^. Sage plants exposed to the variation of temperature followed by 4 °C treatment had an increase in g_s_ from 6.80 (25 °C) to 13.81 (50 °C) mmol m^−2^ s^−1^, followed by a decrease to 4.29 (50 °C) mmol m^−2^ s^−1^.

As can be seen in Figure 1 and Figure 2, a decrease in photosynthetic parameters, both in the net assimilation rate and stomatal conductance to water vapor, can be perceived for both sage and basil plants. 

### 2.2. The Influence of the Variation of Temperatures on the Volatile Organic Compounds Emission

Two green leaf volatile compounds (GLV) have been identified in the blend of volatile emissions from basil and sage plants exposed to different temperatures (Figure 3). The GLVs identified were 3-methyl-pentane and 1-hexanol. The green leaves’ volatiles emission for basil plants exposed only to the temperature variation were 0.05 nmol m^−2^ s^−1^ for 25 °C. With the temperature increase, a maximum of 0.35 nmol m^−2^ s^−1^ can be noticed at 40 °C, followed by a rapid fall of 0.04 nmol m^−2^ s^−1^ at 60 °C. 

For basil plants exposed to the cold followed by temperature variation, a similar pattern was found; a modest rise in GLV was detected from 0.36 (25 °C) to 0.73 (40 °C) nmol m^−2^ s^−1^, followed by a reduction of GLV emission until 0.05 nmol m^−2^ s^−1^ (60 °C). When basil plants were exposed to temperature variation followed by 4 °C, GLV emissions increased from 0.23 (25 °C) to 0.30 (35 °C) nmol m^−2^ s^−1^, and they decreased to 0.11 nmol m^−2^ s^−1^ at 60 °C. As can be seen in Figure 3, for sage plants, the same shape of the graph can be noticed: an increase from 0.02 (25 °C) to 0.44 (45 °C) nmol m^−2^ s^−1^, followed by a decrease to 0.07 nmol m^−2^ s^−1^ at 60 °C. If 4 °C is applied before the variation of temperature, an increase in GLV can be observed from 0.01(25 °C) to 0.24 (45 °C) nmol m^−2^ s^−1^, followed by a decrease to 0.01 (60 °C) nmol m^−2^ s^−1^. If the variation of temperature is applied before 4 °C, the emissions of GLV increase from 0.17 (25 °C) to 0.28 (45 °C) nmol m^−2^ s^−1^, followed by a decrease to 0.20 (60 °C) nmol m^−2^ s^−1^.

The monoterpenes (MT) identified in the mixture of total emissions were α-thujene, tricyclene, α-pinene, camphene, sabinene, β-pinene, β-phellandrene, 3-carene, terpinolene, *trans* and *cis* β-ocimene, 1,8 cineole, D-limonene, γ-terpinene, 4-carene, *trans*-sabinene hydrate, linalool, 2-bornanone, α-terpineol, and bornyl acetate. As shown in Figure 4, for basil plants subjected to the temperature variation, the total MT emission reached a maximum of 35.32 nmol m^−2^ s^−1^ at 45 °C. In the case of the application of 4 °C followed by temperature variation, a maximum MT emission was noticed at 40 °C, namely at 22.16 nmol m^−2^ s^−1^.

When the temperature variation was applied following 4 °C treatment, the highest MT emission was observed at 45 °C, namely 11.43 nmol m^−2^ s^−1^. At 40 °C, the total MT emission from sage plants exposed to temperature variation reached a maximum of 22.38 nmol m^−2^ s^−1^. Upon application of 4 °C followed by temperature variation, the highest MT emission was observed at 35 °C, 22.61 nmol m^−2^ s^−1^. When the temperature variation followed 4 °C, the highest MT emission was observed at 35 °C, namely 23.12 nmol m^−2^ s^−1^.

*Cis*-α-Bergamotene and α caryophyllene were the sesquiterpenes (SQ) identified in the total VOC emission. At 40 °C, the total SQ emission from basil plants exposed to temperature variation reached a maximum of 0.32 nmol m^−2^ s^−1^ (Figure 5).

As can be seen in Figure 5, when 4 °C was applied followed by temperature variation, the maximum SQ emission was observed at 45 °C, namely 0.50 nmol m^−2^ s^−1^. When the temperature variation was followed by 4 °C, the highest SQ emission was observed at 35 °C, namely 0.33 nmol m^−2^ s^−1^. At 50 degrees Celsius, the maximum total SQ emission from sage plants exposed to the temperature variation was 0.16 nmol m^−2^ s^−1^. After applying 4 °C and then varying the temperature, the highest SQ emission was observed at 55 °C, namely 0.37 nmol m^−2^ s^−1^. When the temperature was varied and the 4 °C was applied, the highest SQ emission was observed at 50 °C, namely 0.28 nmol m^−2^ s^−1^.

### 2.3. The Influence of the Variation of Temperatures on the Photosynthetic Pigments

Chlorophyll *a*’s contents differed in all plants subjected to the first or second treatment (Figure 6). 

In the case of basil and sage plants under single temperature stress, the contents of chlorophylls *a* decrease significantly only for plants treated with more than 40 °C until a minimum of 62.4 ± 2.9 µg/mg DW for basil and 16.2 ± 2.4 µg/mg DW for sage plants treated at 60 °C. The chlorophyll *a* concentration decreased for all consecutive treatments compared with the single treatments. In the case of basil plants subjected to the cold–heat treatment, chlorophyll *a*’s content was significantly higher than those subjected to the heat–cold treatment. In the case of sage plants, a higher quantity of chlorophyll *a* was found in plants subjected to heat–cold treatment when compared with the cold–heat treatment. 

The chlorophyll *b* content of basil plants increased to a maximum of 30.9 ± 2.8 µg/mg DW for plants treated at 25 °C, then decreased for all other temperature treatments (Figure 7). When compared with basil plants under one stress, the successive temperature stress significantly reduced chlorophyll *b* content. There are no significant differences for plants treated with consecutive heat–cold stress when compared to those treated with cold–heat stress. The same behavior has been found in sage plants.

### 2.4. The Influence of the Variation of Temperatures on the Total Phenolic Content—Folin−Ciocalteu Method

As can be seen in Figure 8, the total phenolic content (TPC) varied both for basil and sage plants. For basil plants subjected to the single temperature treatment, the TPC maximum was found for plants treated at 40 °C (41.2 ± 3.4 mg GAE/g DW). In the case of the basil plants under successive stresses of cold-heat., there were no significant differences until 45 °C, followed by a slight decrease in TPC. For plants treated with heat followed by 4 °C, the TPC decreased only for plants treated at more than 55 °C.

No significant differences exist for sage plants treated at different single temperature stress (except for 30 °C). When comparing the variation of single-stress temperatures with a successive cold followed by the variation of temperature, a significant decrease could be seen only after 40 °C. In the case of successive heat–cold stress, there was a significant decrease in TPC only after 50 °C. 

### 2.5. The Influence of the Variation of Temperatures on Total Flavonoid Content

As can be seen in Figure 9, the flavonoid content (FC) is quite different for both basil and sage plants.

In the case of basil plants subjected to the single temperature treatment, the optimum temperature for FC was 35 °C (5.56 ± 0.18 mg rutin equivalents/gDW), and a significant decrease has been found only after 45 °C. The highest amount of FC for plants under the successive coldheat treatment was 3.93 ± 0.16 mg rutin equivalents/gDW at 4 + 40 °C. The same behavior has been found for successive heat–cold stress plants, but the highest FC has been found at 30 °C.

For sage plants, the FC was not significantly different for plants treated with successive cold–heat or heat–cold. The highest FC was obtained for plants treated at 40 °C (11.91 ± 0.96 mg rutin equivalents/gDW) for the single temperature treatment, and at 30 °C followed by 4 °C for the consecutive treatment (13.35 ± 1.18 mg rutin equivalents/gDW).

## 3. Discussion

As a result of global warming, extreme temperatures threaten plant quality and yield. Extreme temperatures modify the physiological functions ranging from cellular to organ level. Changes in plant photosynthetic parameters [27], biomass [1,49], enzyme activities [48,49], chlorophyll content [50], mineral uptake [49], antioxidant defense systems [51], total phenolic content [52], antioxidant activity [52], and essential oils [45] were observed. 

Basil and sage plants were exposed to a temperature ramp from 25 to 60 °C in 5-degree increments with and without exposure to cold stress (4 °C). The net photosynthetic rates of basil plants subjected only to one variation of temperature had the highest values at 25 °C, then decreased drastically when 60 °C treatment was applied. At the same treatment, the stomatal conductance to water vapor of basil plants had a slight increase followed by a drastically decreased value when the 60 °C treatment was applied. These findings agree with other studies that have shown that the best growth of sweet basil plants occurrs between 25–30 °C [28,45]. Due to the high temperature, the relative growth rate and net assimilation rate increased significantly in pearl millet but slightly decreased in maize [27]. When comparing the pearl millet, maize, and the plants taken into this study, the net assimilation rate of basil and sage plants also decreases similar to in maize plants [27]. 

The net assimilation rate and stomatal conductance of sage plants exposed to the first treatment have slightly increased at 30 °C, for net assimilation rate and 35 °C for stomatal conductance, followed by a drastic decrease.

When the antagonist heat and/or cold stress is applied both for basil and sage plants, the photosynthetic parameter follows the same proximate pattern, and their value decreases. Cold stress influences photosynthesis and CO_2_ fixation [53]. Some authors report that freezing temperatures degrade bio-membranes due to their extreme inelasticity, resulting in an increase in membrane permeability and a decrease in the activity of membrane-bound ion pumps [38,54], as well as a drastically reduced cellular and chloroplast membrane fluidity [53], all of which contribute to a dramatic decrease in leaf photosynthetic activities [38,54]; cellular and chloroplast membrane fluidity is drastically reduced. Nevertheless, it is known that photosynthesis is sensitive to temperatures well below the freezing point (cold stress) [53]. In two winter wheat cultivars, leaf photosynthesis was significantly inhibited under cold stress, and a decline in net photosynthetic rate and stomatal conductance in seedlings was observed by other authors [55]. Heat stress causes damage to primary photosynthesis processes, specifically PSII and the oxygen-evolving complex, resulting nb an increased membrane fluidity. In our study, even a moderate heat stress, 35 °C, significantly decreased the assimilation rate (Figure 1), possibly reflecting the reduced proton gradient and reduced ATP synthesis [56], as well as decreased stomatal conductance (Figure 2).

The emission of green leaf volatiles, namely 3-methyl-pentane and 1-hexanol, of basil plants increases until reaching a maximum point at 40 °C, then decreases at 60 °C. For basil plants exposed to cold followed by temperature variation, the emissions are higher than for the plants subjected only to the temperature variation. The maximum point reached was 40 °C. When the variation of temperature is applied followed by 4°C, the emission of GLV is lower than the two other treatments. For sage plants, the maximum point of emission of green leaves volatiles is around 45 °C for plants subjected to the variation of temperature; for the cold followed by heat treatment, the maximum emission of GLV is 45 °C, and like in the case of basil plants treated with 4 °C followed by the variation of temperature, the emission of GLV is approximately constant, with only a minor variation. Some studies regarding Norway spruce suggest that GLV increases exponentially with increasing temperature [34].

Because of the absence of reliable physicochemical parameters for specific monoterpenes, little is known about the emission kinetics and storage capacity in leaf gas, liquid, and lipid phases, as well as the thermal protection capability of various monoterpenes [57]. Temperature causes terpene emissions from storage pools to scale rapidly [58]. The monoterpenes identified in the mixture of VOCs in this present study were α-thujene, tricyclene, α-pinene, camphene, sabinene, β-pinene, β-phellandrene, 3-carene, terpinolene, *trans* and *cis* β-ocimene, 1,8 cineole, D-limonene, γ-terpinene, 4-carene, *trans*-sabinene hydrate, linalool, 2-bornanone, α-terpineol, and bornyl acetate. The monoterpene emission increases to thirty times that of normal in basil plants subjected to the variation of temperature, and only eleven times in the case of sage plants. Other authors found that the emission composition of basil mono and sesquiterpenes is made of α-pinene, *trans*-ocimene, D-limonene, *cis*-ocimene, linalool, β-caryophyllene, and *trans*-bergamotene [59]; this fact is in good agreement to our present study.

Sesquiterpene emissions responded similarly to monoterpene emissions in our study, increasing following heat and cold stress. In this present study, most volatile organic compounds were strongly correlated with net assimilation rate and stomatal conductance. In the present study, the composition of emitted mono- and sesquiterpenes were similar between plants subjected only to the variation of temperature and plants submitted to the second treatment (cold or heat). 

The chlorophyll content of basil and sage plants had a maximum at different temperatures between 30–40 °C for all treatments. This fact is due possibly to the optimal growing condition of those plants [45]. Some authors found that basil leaves subjected to low temperatures did not affect chlorophyll concentrations, but they did raise the chlorophyll *a*/*b* ratio in basil leaves [60]. Indeed, in our case, the ratio between chlorophyll *a*/*b* varies between 1.6 and 6.4 for basil plants and between 1.5 and 3.5 in sage plants. There is a lack of research articles regarding *Salvia officinalis* exposure to high temperatures. Only a few studies can be found in the literature. Namely, a reduction of the total chlorophyll content was observed, from 50.93 to 31.92 after 3 days after exposure to high temperature stress, 55 °C and 80% humidity for 30 min [30].

For the total phenolic and flavonoid content of basil and sage plants, the maximum quantity found in the leaves that were exposed to different temperatures was 25–40 °C. Some authors found that the exposure to cold temperatures caused a boost of ascorbic acid in basil cultivars. Such behavior could be explained by the priming effects of cold or heat which determine the increase in plant’s thermotolerance. Still, in the case of lettuce basil leaves, the total phenolic content was enhanced significantly [60]. 

Developing crop plants with increased thermotolerance through a variety of genetic approaches can mitigate the devastating effects of thermal stress. For this reason, a comprehensive understanding of the physiological responses of plants to temperature, the mechanisms of thermal tolerance, and potential strategies for enhancing crop thermotolerance are essential. The results presented here provide a comprehensive evaluation on the effects of extreme temperature on photosynthetic parameters, volatile organic compounds, chlorophyll content, and total phenolic and flavonoid content of basil and sage plants.

## 4. Materials and Methods

### 4.1. Plant Material

Experiments were carried out in 2020 using six-week-old plants of basil (*Ocimum basilicum)* (lot 1GRS. 7A/349702, Horti Tops, Holland Farming Agro, Bucuresti, Romania) and sage (*Salvia officinalis*) (lot 424F0503, Agrosem, Alba, Romania). The seeds were sawn in 0.5 L plastic pots filled with commercial garden soil and grown in growth chambers under controlled conditions of light (1000 µmol m^−2^ s^−1^), light period 12/12 h, day/night temperatures (25/22 °C), and relative humidity (60–70%), and were watered daily to field capacity.

Heat stress was induced by exposing basil and sage plants to different temperatures of 25, 30, 35, 40, 45, 50, 55, and 60 °C for 5 min (control plants for every temperature), as depicted in Figure 10. Basil and sage plants were immersed in distilled water at given temperatures. Then the plants were air-dried for 15 min. For heat stress followed by cold temperatures, the plants were then subjected to 4 °C for 30 min using a refrigerator with preset temperatures (var. temperature—4 degrees) [38]. For cold followed by heat stress, plants were subjected to 4 °C for 30 min (using a refrigerator), followed by a set of high temperatures (25, 30, 35, 40, 45, 50, 55, and 60 °C) for 5 min, as described above (4 degrees—var. temperature). We used in all experiments the 3rd pear of leaves attached to the plants. The experiments have been carried out under the 1000 µmol m^−2^ s^−1^ light intensity. All experiments have been repeated three times with different plants. The fresh weight (FW) of the plants was determined immediately after the leaf was detached. The same leaf has been dried for 72 hours at 70 °C and weighed to determine the dry weight (DW).

### 4.2. Photosynthetic Measurements

After stabilization following the stress application, the leaves subjected to thermal stress were enclosed in a portable gas exchange system GFS-3000 (Waltz, Effeltrich, Germany) to determine the photosynthetic parameters in a method described before [12], Steady-state values of net assimilation rates (A) and stomatal conductance to water vapor (g_s_) were calculated as described before [61].

### 4.3. Volatile Sampling and GC–MS Analyses

Via the outlet of the gas-exchange cuvette at a flow rate of 200 mL min^−1^ for 20 min with a constant flow air sample pump 210-1003 MTX (SKC Inc., Houston, TX, USA), volatile organic compounds (VOC) were sampled using the procedure described before [12]. A Shimadzu TD20 automated cartridge desorber integrated with a Shimadzu 2010 Plus GC–MS instrument (Shimadzu Corporation, Kyoto, Japan) was used to analyze the adsorbent cartridges. The volatile emission rates were calculated as described before [61].

### 4.4. Chromatographic Analysis of Photosynthetic Pigments

Basil and sage leaf samples of 4 cm^2^ were frozen in liquid nitrogen. To extract the photosynthetic pigments, ice-cold acetone (70%) was used, as in the procedure described before [62]. Each extraction was performed in three parallel samples. Chlorophyll *a* and chlorophyll *b* were evaluated using the UHPLC (NEXERA8030, Shimadzu, Tokyo, Japan) equipped with a diode array detector (DAD). 

### 4.5. Total Phenolic Content—Folin− Ciocalteu Method

In order to determine the total phenolic content, methanolic extracts were obtained using fresh leaves 1:10 (*w/v*) in 60% methanol by maceration for 7 days at +4 °C. After maceration, the obtained extracts were filtered using a 0.45 µm PTFE (polytetrafluoroethylene) membrane. The procedure described before evaluated total phenolic content according to the Folin–Ciocalteu method with slight modifications [63].

### 4.6. Flavonoid Content Analysis

The methanolic extracts obtained previously were used to determine the flavonoid content. To determine flavonoid content, 0.250 mL samples were mixed with 1.250 mL of sodium acetate (100 g/L), 0.750 mL of aluminum chloride (25 g/L), and 0.250 mL of water. The absorbance was measured after 15 min, at 434 nm against a blank sample prepared in the same conditions, using a double-beam UV-VIS spectrophotometer (Specord 200, Analytik Jena Inc., Jena, Germany), according to the procedure described before [12,62]. All analyses were performed in triplicate, and the results were reported as mean.

### 4.7. Statistical Analysis and Data Handling

Treatment effects were studied by linear or non-linear regressions whenever pertinent. Means among the treatments were also compared by one-way ANOVA followed by the Tukey’s multiple comparisons tests using GraphPad Prism version 9.3.1 for Windows (GraphPad Software, San Diego, CA, USA). All statistical tests were considered significant at *p* < 0.05. Data sharing different letters are significantly different (*p* < 0.05), while data sharing the same letters are not significantly different (*p* > 0.05).

## 5. Conclusions

Antagonistic stressors influence the plant’s parameters differently and result in higher stress effects than only one stressor. The assimilation rate decreases, and stomata opening is suppressed to cope with different high temperatures, and the VOCs have a maximum emission point depending on the plant species. On the other hand, one stress could prime the plant’s response, regarding its total phenolic compounds and flavonoid contents.

## Figures and Tables

**Figure 1 plants-11-01806-f001:**
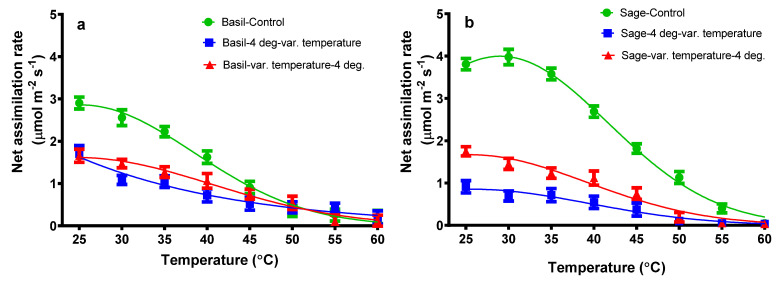
The net assimilation rate from basil plants (**a**) and sage plants (**b**) plants subjected to different temperatures. The values are averages of three individual measurements.

**Figure 2 plants-11-01806-f002:**
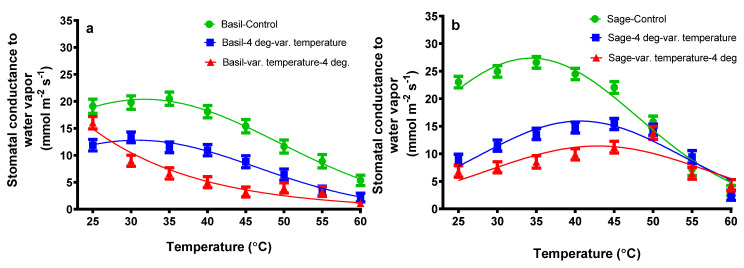
The stomatal conductance to water vapor from basil plants (**a**) and sage plants (**b**) subjected to different temperatures. The values are averages of three independent measurements.

**Figure 3 plants-11-01806-f003:**
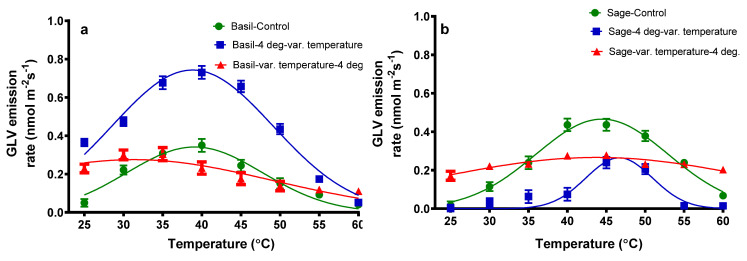
The emission rate of green leaf volatile compounds (GLV) from basil plants (**a**) and sage plants (**b**); plants were subjected to different temperatures.

**Figure 4 plants-11-01806-f004:**
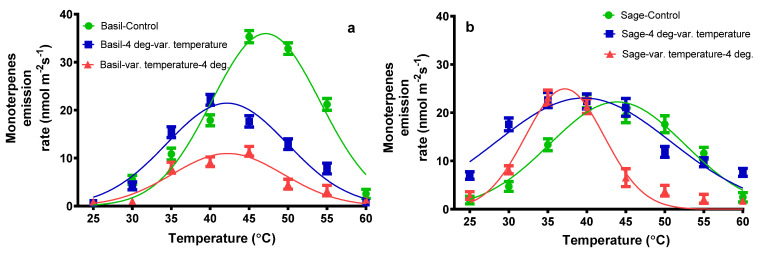
The monoterpenes’ emission rate from basil plants (**a**) and sage plants (**b**); plants were subjected to different temperatures.

**Figure 5 plants-11-01806-f005:**
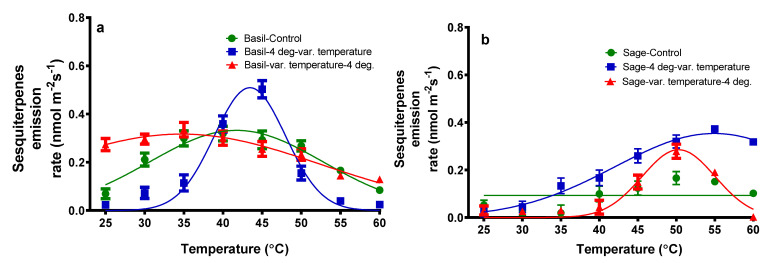
The sesquiterpenes emission rate from basil plants (**a**) and sage plants (**b**); plants were subjected to different temperatures.

**Figure 6 plants-11-01806-f006:**
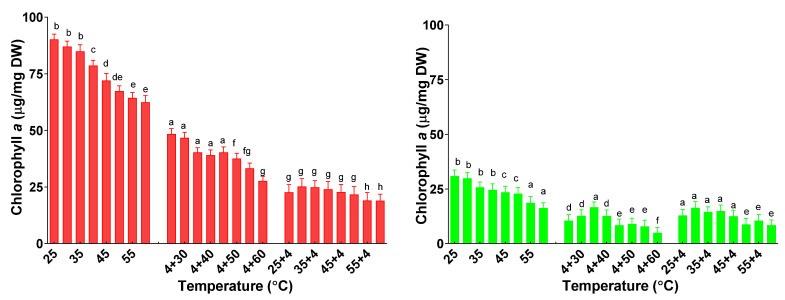
The content of chlorophyll *a* from basil plants (red) and sage plants (green) was subjected to different treatment temperatures. Data sharing different letters are significantly different (*p* < 0.05), while data sharing the same letters are not significantly different (*p* > 0.05).

**Figure 7 plants-11-01806-f007:**
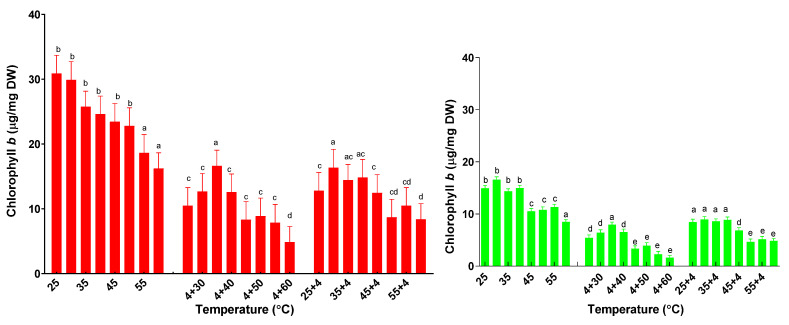
The content of chlorophyll *b* from basil plants (red) and sage plants (green) plants subjected to different temperatures. Data sharing different letters are significantly different (*p* < 0.05), while data sharing the same letters are not significantly different (*p* > 0.05).

**Figure 8 plants-11-01806-f008:**
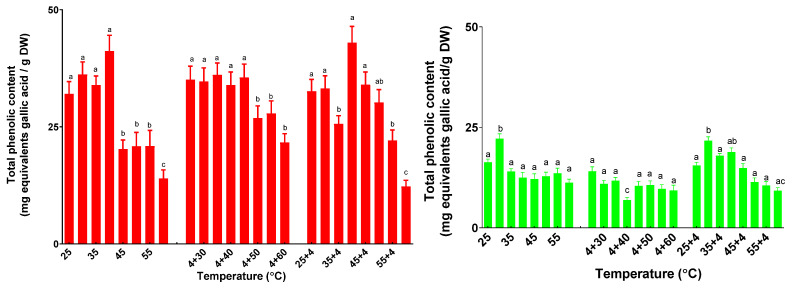
The total phenolic content of basil plants (red) and sage plants (green) plants subjected to different temperatures. Data sharing different letters are significantly different (*p* < 0.05), while data sharing the same letters are not significantly different (*p* > 0.05).

**Figure 9 plants-11-01806-f009:**
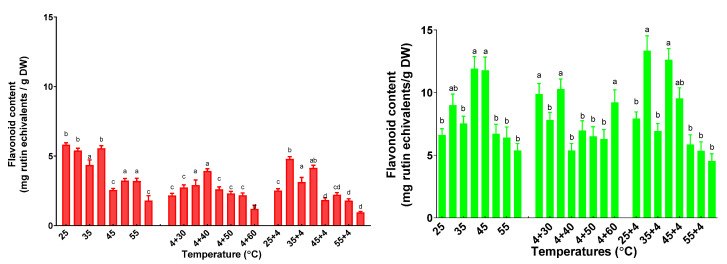
The flavonoid content of basil plants (red) and sage plants (green) plants subjected to different temperatures. Data sharing different letters are significantly different (*p* < 0.05), while data sharing the same letters are not significantly different (*p* > 0.05).

**Figure 10 plants-11-01806-f010:**
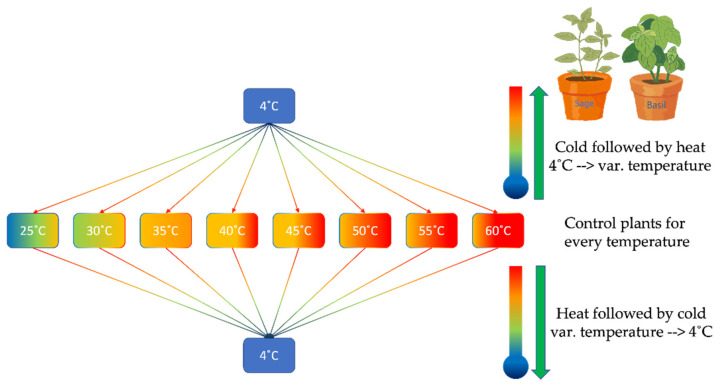
Experimental plan.

## Data Availability

Not applicable.

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
