# Peer review of "Antagonist Temperature Variation Affects the Photosynthetic Parameters and Secondary Metabolites of Ocimum basilicum L. and Salvia officinalis L."

_plants, 2022, doi:10.3390/plants11141806_

Round 1

Reviewer 1 Report

The reviewed paper entitled “Ocimum basilicum L. and Salvia officinalis L. responses to antagonist variation of temperatures” by Copolovici et al. provides data regarding the response of basil and sage plants to various temperatures in terms of emissions of volatiles and other metabolites. These results are of interest and should be published. However, the manuscript requires major corrections before acceptance.

 The reviewer's concerns are listed below:

 Major points:

 1) In the reviewer's opinion the manuscript title is too general and should describe the main findings of the presented research

2) The description of the experimental design is insufficient. The following information is missing: 1) the number of independent experiments; the number of plants used for each variant; 2) the method (if any) for leaves selection for each analytical method; 3) the temperature stress was applied in the presence or absence of light?

Although Authors cited the reference 38 with detailed information about leaf sampling, etc., the more detailed information in the present manuscript is needed. Also in the current manuscript, there is no information that heat treatments were done on detached leaves (according to ref. 38). I am not sure about the low-temperature stress conditions in the current manuscript - in ref 38 the low temperature was applied to the whole plants. In this manuscript we have a combination of low and high temperatures and a precise description of the experimental design is needed. This is essential for proper interpretation of the data. Also, the used experimental design is a combination of temperature and wounding stress, this should be discussed in the manuscript.

3) There is no presentation of the statistical significance of the results. Please, provide appropriate symbols on the figures to present the statistical significance. In many places in the manuscript, the Authors use the phrases ‘slight increase/decrease’. Without statistical significance confirmation, this might be an overstatement.

4) The data for chlorophyll content (Figures 6 and 7) during various temperature treatments look bizarre. It is impossible to observe such an increase in chlorophyll content (in some cases 50 or even near 100%) in the duration of the experiment. The probable explanation of such results is the quantitation of chlorophyll per mg of the leaf fresh weight. Detaching of the leaves in combination with increased/decreased temperature will highly influence the water content in the leaf. Consequently, the actual tissue weight will be highly variable between treatments. The other explanation is sampling leaves of extremely different ages. It should be clarified. Recalculation of the data to chlorophyll content per leaf area (or per mg of dry weight) should help. Also, the calculation of chl a to chl b ratio might be helpful in the interpretation of the temperature impact.

Moreover, in figures 6 and 7 there is an additional point labeled ‘4’. I suppose that is the treatment only with low temperature but there is no description in the manuscript of this treatment. It should be clarified since the presented data for point ‘4’ are highly different compared to other experimental points.

The calculation of the phenolic and flavonoid content per fresh weight (figures 8 and 9) is probably also disturbed due to the reasons stated above.

After verification of the correctness of the data, the rewriting of the abstract, conclusions, and other relevant manuscript sections might be needed.

5) Discussion of the results might be improved. In lines, 321-326 Authors wrote about a strong correlation between volatile compound emission and assimilation rate. It will be good to prepare a summarizing graph showing these correlations. Also, the different temperature maxima for emission of analyzed volatiles in both species and possible reasons should be discussed.

Minor issues:

Authors use two ways to describe control plants: ‘control conditions’ and ‘plants subjected only to temperature variation’. This might be confusing for the readers

Line 103: ‘to 3.98 (35 °C)’; it should be 30 °C

Lines 118-121: some of the data were already described in lines 102-104

Figure 3: use the same Y-axis scale on both panels

Figures 1-9: Symbols used for ‘micro’ and ‘degree’ are incorrectly displayed

Line 357: It should be ‘Figure 10’

Lines 381-382: use italic for ‘a’ and ‘b’ in chlorophyll names. There is no data about b-carotene content in the manuscript.

Author Response

The reviewed paper entitled “Ocimum basilicum L. and Salvia officinalis L. responses to antagonist variation of temperatures” by Copolovici et al. provides data regarding the response of basil and sage plants to various temperatures in terms of emissions of volatiles and other metabolites. These results are of interest and should be published. However, the manuscript requires major corrections before acceptance.

 The reviewer's concerns are listed below:

 Major points:

Q: 1) In the reviewer's opinion the manuscript title is too general and should describe the main findings of the presented research

A:  The title has been changed.

Q: 2) The description of the experimental design is insufficient. The following information is missing: 1) the number of independent experiments; the number of plants used for each variant; 2) the method (if any) for leaves selection for each analytical method; 3) the temperature stress was applied in the presence or absence of light?

A: We added more information to the experimental section.

Q: Although Authors cited the reference 38 with detailed information about leaf sampling, etc., the more detailed information in the present manuscript is needed. Also in the current manuscript, there is no information that heat treatments were done on detached leaves (according to ref. 38). I am not sure about the low-temperature stress conditions in the current manuscript - in ref 38 the low temperature was applied to the whole plants. In this manuscript we have a combination of low and high temperatures and a precise description of the experimental design is needed. This is essential for proper interpretation of the data. Also, the used experimental design is a combination of temperature and wounding stress, this should be discussed in the manuscript.

A: We added more information in the experimental section.

3) There is no presentation of the statistical significance of the results. Please, provide appropriate symbols on the figures to present the statistical significance. In many places in the manuscript, the Authors use the phrases ‘slight increase/decrease’. Without statistical significance confirmation, this might be an overstatement.

A: We added in all figures the statistical analysis of the data.

4) The data for chlorophyll content (Figures 6 and 7) during various temperature treatments look bizarre. It is impossible to observe such an increase in chlorophyll content (in some cases 50 or even near 100%) in the duration of the experiment. The probable explanation of such results is the quantitation of chlorophyll per mg of the leaf fresh weight. Detaching of the leaves in combination with increased/decreased temperature will highly influence the water content in the leaf. Consequently, the actual tissue weight will be highly variable between treatments. The other explanation is sampling leaves of extremely different ages. It should be clarified. Recalculation of the data to chlorophyll content per leaf area (or per mg of dry weight) should help. Also, the calculation of chl a to chl b ratio might be helpful in the interpretation of the temperature impact.

Moreover, in figures 6 and 7 there is an additional point labeled ‘4’. I suppose that is the treatment only with low temperature but there is no description in the manuscript of this treatment. It should be clarified since the presented data for point ‘4’ are highly different compared to other experimental points.

The calculation of the phenolic and flavonoid content per fresh weight (figures 8 and 9) is probably also disturbed due to the reasons stated above.

A:  We re-check all the data and express the concentration on the dry weight of the plants. Indeed there are fewer differences in the concentration of the compound

After verification of the correctness of the data, the rewriting of the abstract, conclusions, and other relevant manuscript sections might be needed.

A: The sections have been changed.

5) Discussion of the results might be improved. In lines, 321-326 Authors wrote about a strong correlation between volatile compound emission and assimilation rate. It will be good to prepare a summarizing graph showing these correlations. Also, the different temperature maxima for emission of analyzed volatiles in both species and possible reasons should be discussed.

A: We improved the results and discussion sections.

Minor issues:

Q: Authors use two ways to describe control plants: ‘control conditions’ and ‘plants subjected only to temperature variation’. This might be confusing for the readers

A: Corrected

Line 103: ‘to 3.98 (35 °C)’; it should be 30 °C

A: Corrected

Lines 118-121: some of the data were already described in lines 102-104

A: Corrected

Figure 3: use the same Y-axis scale on both panels

A: Corrected

Figures 1-9: Symbols used for ‘micro’ and ‘degree’ are incorrectly displayed

A: Corrected

Line 357: It should be ‘Figure 10’

A: Corrected

Lines 381-382: use italic for ‘a’ and ‘b’ in chlorophyll names. There is no data about b-carotene content in the manuscript.

A: Corrected

Reviewer 2 Report

The study is focusing a relevant issue for a large amount of researchers. Yet, the abstract must be improved, since a clear aims (eventually reporting a testing hypothesis - and not the effect of this ..... on that...) must be included. Besides, the abstract also misses a short conclusion about the major issues under investigation. The introduction furnishes a sound background required to further understand the issues under discussion in the paper (therefore can be accepted in the present form). Materials and methods requires additional information (namely day / night temperatures, photoperiod information and day / night hours. In fact, without this information it is difficult to fully understand the reported effects triggered by different temperature treatments. Results seem robust and clearly support the discussion. Moreover the conclusion is mostly a repetition of the results and should be changed (i.e., reporting a general implication of the results, considering the aims of the study).

Author Response

The study is focusing a relevant issue for a large amount of researchers.

Yet, the abstract must be improved, since a clear aims (eventually reporting a testing hypothesis - and not the effect of this ..... on that...) must be included.

A: We improved the abstract

Besides, the abstract also misses a short conclusion about the major issues under investigation.

A: We rewrite the conclusions

The introduction furnishes a sound background required to further understand the issues under discussion in the paper (therefore can be accepted in the present form).

Materials and methods requires additional information (namely day / night temperatures, photoperiod information and day / night hours.

A: Corrected

In fact, without this information it is difficult to fully understand the reported effects triggered by different temperature treatments.

A: Corrected

Results seem robust and clearly support the discussion.

Moreover the conclusion is mostly a repetition of the results and should be changed (i.e., reporting a general implication of the results, considering the aims of the study).

A: We rewrite the conclusions

Round 2

Reviewer 1 Report

In the revised manuscript Authors made substantial amendments. However, there are still issues that had to be corrected/explained before the acceptance of the manuscript.

1) Firstly, recalculation the data from figures 6-9. In the revised version, the Authors express the concentration on the dry weight of the plants. However, it is impossible that compound content per DW (revised manuscript) was lower than per FW (original manuscript). There is still a mistake in the calculations and it must be corrected. Also, there is no description of dry weight calculation (time and temperature of drying).

2) Secondly, The whole manuscript needs to be checked for consistency with new numerical data in figures 7-9. There are a lot of places where only the numbers were corrected but not the text. Below are some examples:

Lines 20-21.  “Chlorophyll concentration for plants under successive stress decrease for basil plants and increase for sage”. This is incorrect, based on the presented data.

Lines 22-23. The increase of flavonoids and phenolic compounds is questionable – it depends on the point of reference. Please be more specific.

Lines: 243-246. This statement is not true. It was not corrected after recalculation of the data presented in Figure 8.

Lines: 257. This fragment is not true. Please look at the previous comment.

Lines: 263-264. This statement is not true. It was not fully corrected after recalculation of the data presented in Figure 9

Lines: 266-269. This is not true. Please look at the previous comment.

3) In the revised version there is still any reference to the data point labeled ‘4’ on figures 6-9.  It should be corrected or removed from the manuscript.

4) In Material and methods Line 385 the phrase ‘control plants’ was removed. If the Authors decided not to use the term ‘control’ for plants subjected only to increased temperature, the other places in the manuscript should be corrected i.e. line 20, lines 104-106, Figures 1-5 legends, Figure 10.

5) During this second review I have noticed that in some places Authors use ambiguous or convoluted sentences to describe the results:

e.g. “For chlorophyll a, in the case of basil plants subjected to the first treatment, the content was significantly higher than basil plants subjected to the second treatment.” (lines 214-216)

or

“For basil plants, when comparing the variation of temperature with cold followed by the variation of temperature and with the variation of temperature followed by cold, it can be seen that the highest quantity of TPC is obtained in the last treatment” (lines 243-245).

Improving the description of the results would have a positive impact on the reception and understanding of the text by readers. However, I leave the decision to the Authors and the Editor.

Author Response

In the revised manuscript Authors made substantial amendments. However, there are still issues that had to be corrected/explained before the acceptance of the manuscript.

Q1) Firstly, recalculation the data from figures 6-9. In the revised version, the Authors express the concentration on the dry weight of the plants. However, it is impossible that compound content per DW (revised manuscript) was lower than per FW (original manuscript). There is still a mistake in the calculations and it must be corrected. Also, there is no description of dry weight calculation (time and temperature of drying).

A: We re-calculate all the data as the reviewer suggested. Indeed there has been a mistake in calculations. Now the data has been revised, and they are correct.

Q2) Secondly, The whole manuscript needs to be checked for consistency with new numerical data in figures 7-9. There are a lot of places where only the numbers were corrected but not the text. Below are some examples:

Lines 20-21. “Chlorophyll concentration for plants under successive stress decrease for basil plants and increase for sage”. This is incorrect, based on the presented data.

Lines 22-23. The increase of flavonoids and phenolic compounds is questionable – it depends on the point of reference. Please be more specific.

Lines: 243-246. This statement is not true. It was not corrected after recalculation of the data presented in Figure 8.

Lines: 257. This fragment is not true. Please look at the previous comment.

Lines: 263-264. This statement is not true. It was not fully corrected after recalculation of the data presented in Figure 9

Lines: 266-269. This is not true. Please look at the previous comment.

A: In order to have consistency, all results part (regarding figures 6-9) have been re-written.

3) In the revised version there is still any reference to the data point labeled ‘4’ on figures 6-9. It should be corrected or removed from the manuscript.

A: In order to have consistency with figures 1-5, we decide to remove point “4” from figures 6-9.

4) In Material and methods Line 385 the phrase ‘control plants’ was removed. If the Authors decided not to use the term ‘control’ for plants subjected only to increased temperature, the other places in the manuscript should be corrected i.e. line 20, lines 104-106, Figures 1-5 legends, Figure 10.

A: We added the control plants in M&M.

5) During this second review I have noticed that in some places Authors use ambiguous or convoluted sentences to describe the results:

e.g. “For chlorophyll a, in the case of basil plants subjected to the first treatment, the content was significantly higher than basil plants subjected to the second treatment.” (lines 214-216)

or

“For basil plants, when comparing the variation of temperature with cold followed by the variation of temperature and with the variation of temperature followed by cold, it can be seen that the highest quantity of TPC is obtained in the last treatment” (lines 243-245).

Improving the description of the results would have a positive impact on the reception and understanding of the text by readers. However, I leave the decision to the Authors and the Editor.

A: In order to have consistency, all results part (regarding figures 6-9) have been re-written.

We want to thank you for all your help and suggestions in order to improve the article.

Round 3

Reviewer 1 Report

I am satisfied with the corrections to the manuscript made by the Authors. I only have a few minor comments:

Line 249: I think, the ‘cold-heat’ should be here instead of ‘heat-cold’.

Line 251: I think, ‘plants treated with heat followed 4 ⁰C’ (or substitute) should  be here

Line 279-280: please check the consistency with figure 9

 line 402: it should be ‘12/12 h day/night’